**Review**

# Molecular insights into diverse heat hormesis regimens in *Caenorhabditis elegans*

Hsin-Yun Chang, Charles L. Heinke 🄳 , Siu Sylvia Lee*

Department of Molecular Biology and Genetics, Cornell University, Ithaca, NY 14853, United States

*Corresponding author: Department of Molecular Biology and Genetics, Cornell University, 339 Biotechnology Building, 215 Tower Rd, Ithaca, NY 14853, United States. Email: sylvia.lee@cornell.edu

Heat hormesis describes a biphasic, dose-dependent response in which low levels of heat stress induce beneficial effects, such as enhanced lifespan and stress resilience. This phenomenon is commonly studied in *Caenorhabditis elegans* using regimens that involve mild heat stress priming, followed by a recovery period and subsequent stress challenge or lifespan measurement. The concept is conserved across species and has gained renewed interest due to its potential relevance to therapeutic strategies, including sauna-based wellness practices. This review synthesizes phenotypic and molecular findings from *C. elegans* heat hormesis studies, organizing them by regimen type, defined by temperature, duration, and developmental stage of priming. This structure enables comparisons across regimens, revealing both shared and distinct physiological outcomes and mechanistic responses. We highlight current knowledge gaps and discuss considerations for future experimental design, including more consistent control of key variables, to support efforts in identifying optimized conditions with potential translational relevance for health and therapeutic applications. Lastly, we draw from heat hormesis studies performed in mammals to compare methodologies and emphasize conserved mechanisms.

Keywords: heat hormesis; thermotolerance; *C. elegans*; stress adaptation; aging; longevity

## Introduction

Hormesis is a widely observed biological phenomenon across metazoans, describing an adaptive response in which exposure to low levels of stress induces beneficial effects, while higher levels of stress become detrimental (Calabrese and Baldwin 2002; Mattson 2008; Calabrese 2014). Hormesis has been increasingly recognized as a fundamental adaptive response, where mild stress exposure acts as a preconditioning stimulus, activating protective mechanisms that enhance resilience against subsequent, more severe stress challenges (Gems and Partridge 2008; Calabrese and Mattson 2017) (Fig. 1). Given the broad definitions and interpretations of hormesis, this review specifically refers to "hormesis" or "hormetic effect/response" as beneficial, protective adaptations resulting from exposure to mild stressors. These beneficial effects include enhanced stress resilience, increased lifespan, and improved healthspan. While strong correlations exist among these phenotypes, they may also arise through independent mechanisms and do not always co-occur. Hormesis can be induced by a wide range of stressors, extensively documented in the literature, including temperature, radiation, heavy metals, oxidative stress, dietary modifications, and more (Cypser and Johnson 2002; Gems and Partridge 2008; Mattson 2008; Calabrese 2014). A key feature of hormetic responses is "cross-protection/cross-tolerance," where adaptation to one stressor confers resistance to multiple other stressors (Calabrese et al. 2007; Kishimoto et al. 2017; Berry and López-Martínez 2020). This phenomenon highlights the systemic nature of stress responses and their potential implications for health and disease prevention.

The significance of hormesis has gained increasing attention in both biomedical research and therapeutic applications. For example, ischemic preconditioning, where brief, controlled periods of ischemia induce tolerance to subsequent ischemic events, has been shown to reduce the severity of ischemic heart disease, cerebral ischemia, and stroke (Cohen and Downey 2015; Yang et al. 2022). More recently, hormesis has gained mainstream popularity due to its association with extending healthspan and potentially slowing the ageing process (Li et al. 2019; Calabrese et al. 2024). Popular wellness practices such as heat therapy, cold exposures, and intermittent fasting are thought to engage hormetic mechanisms to improve health and potentially mitigate ageing-related diseases (Longo and Mattson 2014; Most et al. 2017; Laukkanen et al. 2018; Patrick and Johnson 2021). A recent bibliometric analysis provided a comprehensive review of hormesis research trends, highlighting its expanding relevance in disease therapeutics and longevity interventions (Wan et al. 2024).

Despite extensive documentation of hormetic phenotypes across species, the molecular mechanisms underlying hormesis remain complex and context-dependent. These mechanisms vary depending on multiple factors, including the type and intensity of the stressor, the developmental stage at which exposure occurs, and the specific protective outcomes monitored. Research has linked hormetic effects to cellular signaling pathways, transcriptional alterations, epigenetic modifications, and protein maintenance (Gems and Partridge 2008; Mattson 2008; Wan et al. 2021; Perrone and D'Angelo 2025). Dynamic stress-signaling networks are the key to counteracting damage through changes in gene regulation. Epigenetic modifications likely contribute to

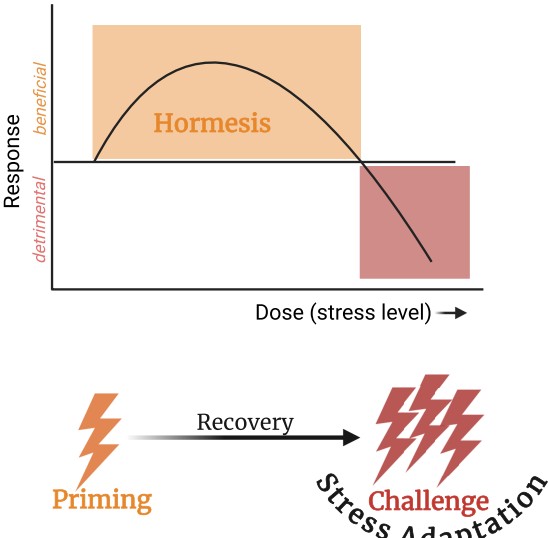

**Fig. 1.** Biphasic dose–response curve illustrating hormesis and its connection to stress intervention regimens. The upper panel depicts an inverted U-shaped relationship between dose (stress level) and physiological response. Low-dose stress induces a beneficial "hormetic" response (orange zone), whereas high-dose stress results in a detrimental response (red zone). The lower panel illustrates a common experimental regimen used in *Caenorhabditis elegans* hormesis studies, consisting of a mild stress priming, followed by a recovery period, and then a high-dose stress challenge. This approach is used to evaluate whether priming induces beneficial hormetic effects, such as enhancing stress adaptation. Created in BioRender. Lee, S. (2026) https://BioRender.com/sqrxa3y.

long-term "memory-like" adaptations that sustain and transmit the protective effects of hormesis across an organism's lifespan and even across generations. Protective memory can also be at the protein level by maintaining long-term proteostasis. These mechanisms will be elaborated within the context of a specific type of hormesis regimen in the following sections.

A well-documented form of hormesis, "heat hormesis," where exposure to thermal stress induces beneficial effects, has been extensively studied across multiple species, including worms, flies, yeast, mice, and human cells, demonstrating its conserved and systemic impact on biological function (Shama et al. 1998; Le Bourg et al. 2001; Verbeke et al. 2002; Yokoyama et al. 2002; Hercus et al. 2003; Rattan 2005; Cypser et al. 2006; Rattan et al. 2009; Mane et al. 2018). Experimental evidence has demonstrated that transient mild heat exposure can extend lifespan, suggesting heat hormesis has broad implications for ageing biology (Le Bourg et al. 2001; Hercus et al. 2003; Lagisz et al. 2013). In mammalian models and human cells, hormetic heat stress has been linked to improved wound healing, enhanced angiogenesis, and cellular differentiation, further emphasizing its therapeutic potential (Verbeke et al. 2002; Rattan et al. 2009). In addition to laboratory models, sauna therapy, a form of controlled heat exposure, has been associated with reduced risk of cardiovascular and neurodegenerative diseases and improved physical fitness, highlighting its potential role as a lifestyle intervention for promoting long-term health (Laukkanen et al. 2018; Patrick and Johnson 2021). Moreover, evidence suggests that heat hormesis may extend beyond individual organisms, with transgenerational benefits (Das et al. 2021; Wan et al. 2021).

This review aims to connect specific heat stress regimens with their phenotypic outcomes and underlying molecular mechanisms. The beneficial effects of heat hormesis have been widely observed across various mild heat exposure conditions, which differ in terms of temperature, duration, and timing. However, the molecular mechanisms driving these effects may differ depending on the specific regimen. Our goal is to clarify how specific heat regimens shape biological responses at both phenotypic and mechanistic levels, offering insight into the potential for tailored therapeutic applications of hormesis. This review largely focuses on *C. elegans*, a model organism typically maintained at 15 to 20 °C in laboratory conditions, although we have included a short comparison with mammalian studies to highlight conserved mechanisms. Future comparative studies across species will be essential for uncovering conserved and divergent features of heat hormesis and for strengthening its relevance for translational applications. While there is extensive and exciting research on the immediate molecular effects of heat stress, the scope of this review will be limited to longer-term physiological hormetic benefits such as lifespan extension and increased stress resilience.

To provide an understanding of how timing, stress intensity, and developmental stage influence the outcomes of heat-induced hormesis, we organize our discussion of heat hormesis regimens into 5 categories:

1) Single heat stress priming in early adulthood—prolonged
2) Single heat stress priming in early adulthood—short
3) Repeated heat stress priming in adulthood
4) Priming in late adulthood
5) Chronic heat exposure during development

An overview of the reviewed studies and key findings is provided in Table 1.

## Single heat stress priming in early adulthood - prolonged

This section covers studies involving single prolonged heat stress priming on day 1 of adulthood, characterized by lower temperatures (30 °C) and longer durations (3–24 h) compared to the short priming regimens discussed in Section 2.

### *Priming duration threshold altered in long-lived mutant*

One of the earliest studies to identify heat hormesis phenotypes in *C. elegans* dates back to 1995, shortly after the discovery of several longevity-associated mutant genes that revolutionized ageing research (Johnson 1990; Kenyon et al. 1993; Kenyon 2005). Shortly after their discovery, it was noticed that several of these long-lived mutants also exhibited an innate resilience to heat stress. Lithgow et al. tested the effects of a single mild heat exposure (30 °C for 3–24 h) on both wild-type (WT) and the long-lived *age-1*/PI3 K loss-of-function mutant at the 4-d-old stage (approximately 1 d after reaching sexual maturity, referred to as day 1 adult hereafter), which were otherwise maintained at 20 °C. They observed similar enhancement in thermotolerance (an increase in survival after intense heat stress, sometimes referred to as "thermoresistance" in the literature) after a 12-h recovery and lifespan extension in both WT and *age-1* mutants. Specifically, a 3-h preconditioning increased thermotolerance in WT but not in the *age-1* mutant, whereas longer exposures (6 h 45 min and 12.5 h) conferred thermotolerance to both strains. Moreover, preconditioning for 6, 12, and 24 h significantly extended lifespan in both WT and *age-1* mutants, suggesting a potential link between thermotolerance and longevity (Lithgow et al. 1995). This early study provided evidence that transient mild heat stress can extend lifespan in WT and further extend lifespan in the already long-lived

**Table 1.** Overview of reviewed studies on heat hormesis in *Caenorhabditis elegans*.

| Regimen Type | Priming Condition | | | Strain | Observed Physiological Effects | Molecular Insights | Reference |
|---|---|---|---|---|---|---|---|
| | Temperature | Duration | Priming Stage (Timing of Intervention) | | | | |
| 1. Single heat stress priming in early adulthood—Prolonged | 30 °C | 6, 12, 24 h (3), 6.75, 12.5 h | Day 1 adult | WT, age-1(−) WT, age-1(−) | • Extended lifespan<br>• Increased thermotolerance (12-h recovery) | | Lithgow et al. (1995) |
| | | 6 h | | WT | • Extended lifespan | • HSP protein induction | Yokoyama et al. (2002) McColl et al. (2010) |
| | | 6 h | | hsf-1(sy441) | • Extended lifespan | • HSF-1 partially required | Chang (2025) |
| | | 6 h | | WT, glp-1 (ts) | • Increased thermotolerance (12-h recovery) | • Global dose-dependent transcriptomic and chromatin accessibility response<br>• MARS-1/MARS1, SNPC-4/SNAPc, FOS-1/c-Fos, and DPY-27/SMC4 are required | |
| | 25 °C | 18 h | | WT | • Increased thermotolerance<br>• Increased hypoxia resistance<br>• Increased cadmium resistance | • HIF-1 upregulation, conserved in mammals | Treinin et al. (2003) |
| 2. Single heat stress priming in early adulthood—Short | 35 °C | 1, 2 h | Day 1 adult | WT | • Increased thermotolerance (12-h recovery)<br>• Extended lifespan | | Cypser and Johnson (2002) |
| | 35 °C | 2 h | | TJ1060 [fer-15(b26); spe-9(hc88)] daf-12(−), daf-16(−), daf-18(−) | • Extended lifespan<br>• Increased thermotolerance (12-h recovery) | • Regulated by IIS pathway | Cypser and Johnson (2003) |
| | 34 °C | 30 min | | WT | • Protection against heat stroke-induced neuron damage | • HSF-1 induces HSP-16.1 localization to Golgi to maintain Ca$^{2+}$ homeostasis<br>• DAF-16/FOXO is partially required | Kourtis et al. (2012) |
| | 35 °C | 1 h | | WT | • Increased cadmium resistance via neuroprotection | • DAF-16/FOXO and HSF-1 mediated HSP-16.2 induction | Wang et al. (2020) |
| | 36 °C | 30, 60 min | | WT | • Extended lifespan<br>• Increased thermotolerance (3-d recovery)<br>• Reduced PolyQ aggregation | • Autophagy-mediated proteostasis (regulated by HLH-30/TFEB) | Kumsta et al. (2017) |
| | 35 °C | 1 h | | WT | • Increased thermotolerance<br>• Increased cadmium resistance | • ENDU-2 mediates transcriptional reprograming during post-heat stress | Xu et al. (2023) |
| | 35 °C | 1 h | | WT | • Transgenerational<br>• Lifespan extension<br>• Reduction of protein aggregates (α-synuclein) | • DAF-16/FOXO, H3K9me3, and N6-mA modifications mediate the transgenerational inheritance<br>• HSF-1, DAF-16/FOXO, and DAF-12/FXR implement the protective response | Wan et al. (2021) |

**Table 1.** (continued)

| Regimen Type | Priming Condition | | | Strain | Observed Physiological Effects | Molecular Insights | Reference |
|---|---|---|---|---|---|---|---|
| | Temperature | Duration | Priming Stage (Timing of Intervention) | | | | |
| | 34 °C | 5 min | | WT | • Transgenerational <br> • Increased thermotolerance <br> • Reduction of protein aggregates (PolyQ) | • Germline HSF-1-dependent H3K9me2 deposition via MET-2 | Das et al. (2021) |
| 3. Repeated heat stress priming in adulthood | 33 °C | 4 h | Day1 + 5 + 9 + 13 adult | WT | • Greater lifespan extension than single exposure | • HSP induction in early adulthood but is not inducible in older adults | Olsen et al. (2006) |
| | 30 °C | 4 h | Day 1 + 5 + 9 adult | TJ1060 [fer-15(b26); spe-9(hc88)] | | | |
| 4. Priming in late adulthood | 33 °C | 4 h | Day 5 or 9 adult | WT | • Extended lifespan | | Olsen et al. (2006) |
| | 36 °C | 1 h | Day 3 <br> Day 2, 3 or 5 adult | WT | • Extended lifespan <br> • Increased thermotolerance (12-h recovery) | • HSP and autophagy gene induction from Day 1 to 7 of adulthood stage | Kumsta et al. (2017) |
| 5. Chronic heat exposure during development | 25 °C | whole development period | | WT | • Extended lifespan | • DAF-16/FOXO dependent transcriptional changes via TRPA-1 thermosensitive channel | Zhang et al. (2015) |
| | | whole development period | | WT | • Increased pathogen resistance <br> • Increased oxidative stress resistance | • CBP-1 and SWI/SNF-mediated persistent gene induction downstream of PMK-1/MAPK | Zhou et al. (2019) |
| | | 1 d, starting at L4 stage | | WT | • Extended lifespan <br> • Increased resistance to HS, UV, and oxidative stress | • Regulated by HSF-1, DAF-16/FOXO, and HIF-1 <br> • Improved proteostasis via decreased global protein synthesis and enhanced protein maintenance/degradation | Huang et al. (2023) |
| | | 1 d, starting at L4 stage | | WT | • Extended lifespan <br> • Increased thermotolerance | • Transient tsp-1 mRNA induction, mediated by CBP-1, lead to TSP-1 protein multimerization for long-term protection | Jiang et al. (2024) |

This table summarizes key studies examining heat hormesis across different experimental regimens. Regimen types are categorized based on priming conditions (temperature, duration, and priming stage), with associated strain(s), observed physiological effects, and molecular insights included.

*age-1* mutant. This highlights the significant role of environmental stimuli in promoting longevity beyond genetic predispositions.

### Sustained induction of HSPs after priming

The heat shock response (HSR) has been linked to heat hormesis, as mild heat exposure induces the overproduction of heat shock proteins (HSPs), which act as molecular chaperones. These proteins not only repair transient cellular damage but may also confer persistent protection against long-term stress-related damage (Kurapati et al. 2000). Yokoyama et al. applied a mild heat regimen (6 h at 30 °C) to day 1 adults and observed a significant lifespan extension, even though the worms were maintained at 25 °C instead of 20 °C. Additionally, their study provided mechanistic insights by demonstrating that this regimen induces the expression of HSPs, particularly HSP-70F (today known as HSP-6, a component of the mitochondrial unfolded protein response), whose protein levels remained elevated for up to 3 d post-heat exposure (Yokoyama et al. 2002). This finding suggests that inducible HSPs may play a key role in mediating the longevity benefits associated with heat hormesis, although direct causal evidence remains to be established.

### A conditional requirement of HSF-1

McColl et al. investigated the role of Heat Shock Factor 1 (HSF-1), the master transcriptional regulator of HSR, in the enhanced thermotolerance observed in *C. elegans* following the same priming regimen (30 °C for 6 h on day 1 of adulthood, with 12-h recovery)(McColl et al. 2010). Using the *hsf-1 (sy441)* mutant, they found that this allele partially impaired the priming-induced increase in thermotolerance, indicating that HSF-1 contributes to the beneficial effects of heat hormesis. It should be noted that the precise molecular consequences of *hsf-1 (sy441)* mutant remain under investigation. This allele introduces a truncation that removes the C-terminal transactivation domain while retaining the DNA-binding domain, potentially allowing HSF-1 to bind heat shock elements without fully activating target gene expression (Kurapati et al. 2000; Hajdu-Cronin et al. 2004; Kovács et al. 2024), thus possibly acting as a dominant negative mutant. Somewhat contrary to McColl's observation of HSF-1 being partially required for heat hormesis, 2 recent studies assert that interruption of HSF-1 function, either using *hsf-1 (sy441)* or RNAi, fully eliminates hormetic heat priming benefits (Wan et al. 2021; Kovács et al. 2024). However, it should be noted that both studies used 5-fluoro-2′-deoxyuridine (FUDR) to block germline mitosis during their thermotolerance assays of hormesis. Our lab has independently observed a full requirement for HSF-1 in the germline-less *glp-1(ts)* mutant, but only a partial requirement in wild-type N2 worms, in thermotolerance induced by hormetic heat priming (Chang et al. 2025). These observations together suggest that inhibition of germline proliferation renders HSF-1 essential for heat priming-induced thermotolerance, but its role appears less crucial in germline-competent worms. HSF-1 is known to enhance germline proliferation and has germline-specific DNA targets (Edwards et al. 2021). Because ablation of germline stem cells has been directly connected to lifespan extension (Hsin and Kenyon 1999; Arantes-Oliveira et al. 2002), it follows that disruption of HSF-1 could elicit both pro- and anti-survival effects within germline-competent worms.

### HIF-1 required for cross-stress protection

Treinin et al. examined the role of Hypoxia-Inducible Factor 1 (HIF-1) in mediating hormetic benefits, which they also referred to as heat acclimation (AC). They applied a priming regimen of 18 h at 25 °C on day 1 of adulthood and found that treated worms exhibited enhanced thermotolerance upon re-exposure to 35 °C immediately after priming, as well as increased protection against hypoxia and cadmium stress. Mechanistically, analysis of loss-of-function and gain-of-function mutants demonstrated that HIF-1 upregulation is required for these protective effects (Treinin et al. 2003). This study identified HIF-1 as an essential regulator of cross-stress tolerance induced by heat hormesis, a mechanism that is conserved in mammals and will be discussed in later sections.

### Dose-dependent global transcriptomic and chromatin accessibility response to heat hormesis

A recent study from our lab by Chang et al. observed that prolonged priming in early adulthood (30 °C for 6 h on day 1 of adulthood) enhanced thermotolerance in both WT and germline-less *glp-1(ts)* mutant strains. This study provided a comprehensive view of transcriptional and chromatin dynamics throughout the heat hormesis process, including time-series tracking of molecular changes from the initial mild stress priming, through recovery (12 h), and upon subsequent heat shock challenge (35 °C), in both backgrounds (Chang et al. 2025). This study revealed distinct changes in mRNA expression and chromatin accessibility in response to low vs high levels of heat stress, providing molecular evidence supporting dose-dependent stress responses. Additionally, although both mRNA expression and chromatin accessibility largely return to baseline after 12 h of recovery, heat-adapted organisms exhibit significant transcriptional and chromatin differences upon subsequent heat shock. These findings suggest the existence of molecular "memories" that may underlie the physiological benefits observed in stress-adapted organisms.

### MARS-1/MARS1, SNPC-4/SNAPc, FOS-1/c-fos, and DPY-27/SMC4 are required for thermotolerance

Chang et al. conducted RNAi screening of candidate hormesis regulators identified through multiomic analysis, revealing several essential genes for conferring heat priming-induced thermotolerance in both WT and *glp-1(ts)*(Chang et al. 2025). These include MARS-1, an aminoacyl-tRNA synthetase involved in methionine incorporation during protein synthesis; SNPC-4, a subunit of the SNAPc complex required for piRNA biogenesis; FOS-1, a component of the AP-1 pioneer factor that facilitates the binding of other transcription factors (Patrick et al. 2024); and DPY-27, a subunit of the dosage compensation complex that regulates chromosome architecture (Meyer 2022). These findings indicate that heat hormesis engages a broad range of biological pathways, including translation, small RNA regulation, transcriptional control, and chromosome organization. It is likely that different steps of gene expression regulation are engaged to confer appropriate RNA and protein expression changes that help to resolve the stressor, and larger scale chromatin changes provide a longer-term memory for adaptive response.

### Germline-less and long-lived **glp-1(ts)** mutants do not exhibit further lifespan extension upon priming

Chang et al. reported that priming (30 °C for 6 h on day 1 of adulthood) extends lifespan in WT, consistent with previous studies (Lithgow et al. 1995; Yokoyama et al. 2002). However, this effect was not observed in the *glp-1(ts)* mutant, which lacks a functional germline and is already long-lived. The authors proposed that transient disruption of germline development during priming is sufficient to trigger a pro-longevity signal in WT animals.

Supporting this, they observed a reduction in total brood size following priming, with egg production impaired primarily during the priming period and recovering afterward (Chang et al. 2025). Moreover, heat priming induces gene expression changes that substantially overlap with those induced in germline-ablated animals, further supporting the notion that heat priming transiently halts germline proliferation, thereby promoting longevity (Chang et al. 2025). These findings highlight the need to disentangle the contributions of stress-induced signaling and reproductive status in mediating hormesis-associated longevity.

## Single heat stress priming in early adulthood—short

This section covers studies involving single heat stress priming on day 1 of adulthood under acute conditions, characterized by higher temperatures (34–36 °C) and shorter durations (0.5–2 h).

A transient exposure to 35 °C on day 1 of adulthood is another commonly used regimen for studying heat hormesis in *C. elegans*. Cypser and Johnson investigated this regimen in WT worms maintained at 20 °C and found that 1- and 2-h priming at 35 °C significantly increased thermotolerance after a 12-h recovery period, with the effect peaking at 2 h. However, no beneficial effects were observed with 3-h exposure, and longer priming periods resulted in a decline in thermotolerance. Similarly, when using the temperature-sensitive sterile strain TJ1060 [*fer-15(b26); spe-9(hc88)*], a comparable hormetic pattern was observed for lifespan extension. Specifically, 1- and 2-h priming at 35 °C increased lifespan, whereas 3 h showed no effect, and 4 h became detrimental (Butov et al. 2001; Michalski et al. 2001; Cypser and Johnson 2002). These findings highlight the narrow optimal window for heat-induced benefits.

### IIS pathway involvement

Using the optimal regimen (35 °C for 2 h), Cypser and Johnson further found that *daf-12*/FXR, *daf-16*/FOXO, and *daf-18*/PTEN, 3 key components of the insulin/IGF-1-like signaling (IIS) pathway, were required for heat hormesis-induced lifespan extension. However, only *daf-18*, and not *daf-12* or *daf-16*, was fully required for increased thermotolerance (Cypser and Johnson 2003). This study provided direct evidence that dauer-related genes play a critical role in heat hormesis, suggesting that mechanisms beyond HSR contribute to the beneficial effects of mild heat stress. It is well-established that HSF-1 is required for IIS-mediated longevity; further investigation into the interplay between these pathways in regulating heat hormesis could provide deeper insights.

### Protection against neuron damage via $ca^{2+}$ homeostasis

Kourtis et al. (2012) demonstrated that heat hormesis provides protection against heat stroke-induced neuron damage in *C. elegans*. They tested this by priming worms at 34 °C for 30 min during day 1 of adulthood (early egg-laying), 20 min before subjecting them to a 15-min heat stroke challenge at 39 °C. Unlike standard thermotolerance/thermoresistance assays that typically use 35 to 37 °C, heat stroke assays involve more acute and extreme heat stress at 39 °C, mimicking hyperthermia in humans, which causes severe and immediate physiological damage. In worms, heat stroke leads to a sharp increase in mortality, with less than 25% of worms surviving after 18 h. At the cellular level, widespread necrotic cell death, rather than apoptosis or autophagy, was observed across multiple tissues, including dopaminergic neurons, highlighting the devastating effects of excessive heat stress. This study revealed a cytoprotective mechanism induced by heat hormesis that defends against necrotic cell death,

shedding light on a potential conservation of this response in pathological necrosis associated with hyperthermia in humans. Mechanistic investigations identified an HSR-dependent protective pathway downstream of HSF-1, in which HSP-16.1, induced by hormetic heat stress, localizes to the Golgi apparatus. There, it interacts with the $Ca^{2+}/Mn^{2+}$-transporting ATPase PMR-1, helping maintain $Ca^{2+}$ homeostasis under heat stroke conditions. Additionally, DAF-16/FOXO was found to be partially required for heat hormesis-induced protection against heat stroke, consistent with its synergistic role with HSF-1 in facilitating HSP induction.

### Protection against cadmium via DAF-16/FOXO, HSF-1

Wang et al. found that heat hormesis also provides protection against cadmium-induced lethality, a toxic heavy metal that is ubiquitously present in the environment and damages multiple physiological functions, including reproduction. Using a heat hormesis regimen of 35 °C for 1 h on day 1 of adulthood followed by a 1-h recovery period at 20 °C before the cadmium lethality assay, they observed that hormetic heat stress significantly enhanced cadmium resistance. This was evidenced by increased survival rates, reduced bagging formation, and improved intestinal barrier integrity. Furthermore, their study revealed that heat priming leads to an increase in serotonergic neuron size, suggesting that heat hormesis enhances cadmium resistance by protecting neurons, which in turn mitigates cadmium-induced physiological damage. Mechanistically, they demonstrated that cadmium resistance induced by heat hormesis is mediated through the DAF-16/FOXO and HSF-1 pathways via the regulation of HSP-16.2 expression (Wang et al. 2018, 2020). These findings provide insights into the neuroprotective role of heat hormesis in counteracting heavy metal toxicity.

### Involvement of autophagy-mediated proteostasis

Kumsta et al. (2017) linked the protective mechanisms of heat hormesis to an HSF-1-mediated autophagy pathway that functions in parallel with HSR. Using a priming regimen of 30- to 60-min exposure to 36 °C on day 1 of adulthood, they observed lifespan extension and enhanced thermotolerance following a 3-d recovery period at 20 °C. Their study demonstrated that hormetic heat stress induces autophagy, as evidenced by GFP reporter strains tracking autophagy gene expression. Moreover, knocking down autophagy-related genes abolished the heat-induced benefits on both thermotolerance and longevity, establishing autophagy as a crucial mediator of heat hormesis. In addition, they found that HLH-30/TFEB, a conserved transcription factor and a key regulator of autophagy gene expression, translocated to nuclei in response to hormetic heat stress and is also required for the beneficial effects on both thermotolerance and longevity. To explore further, Kumsta et al. demonstrated that heat hormesis also enhances proteostasis through autophagy. Using a PolyQ aggregation model in multiple tissues, they observed a significant reduction in PolyQ puncta aggregation in heat hormesis-treated worms, an effect that was abolished upon autophagy gene knockdown. Moreover, heat hormesis increased the lifespan of neuron- and intestine-specific PolyQ disease models (Kumsta et al. 2017). A follow-up study revealed that the autophagy receptor SQST-1/p62 was essential for priming-induced lifespan and thermotolerance benefits (Kumsta et al. 2019). Interestingly, while not required for basal autophagy in most tissues, SQST-1 was required for priming-induced autophagy in all tissues analyzed. These studies link heat hormesis and autophagy, establishing autophagy as a cytoprotection mechanism, in addition to HSR, in promoting

stress resistance, longevity, and the mitigation of age-related protein aggregation diseases.

### ENDU-2-mediated transcriptional reprogramming

Xu et al. (2023) discovered that transcriptional reprogramming following hormetic heat stress (35 °C for 1 h) is regulated by the RNA-binding protein ENDU-2, specifically during the post-heat stress period, in a manner distinct from the canonical HSR and independent of HSF-1. Through transcriptomic profiling immediately after hormetic heat stress and 4 h post-stress, they revealed that gene expression changes during and after heat exposure were largely distinct. Specifically, gene expression changes initiated during heat stress could be maintained or even further enhanced post-stress, and a distinct set of genes was induced exclusively during the recovery phase. Functionally, hormetic heat stress–induced thermotolerance and cadmium resistance were abolished in the absence of ENDU-2, as well as several ENDU-2-regulated post-heat stress response genes, including *hsp-16.2, zip-10, irg-2,* and *pqm-1*. Mechanistic studies demonstrated that hormetic heat stress promotes ENDU-2 binding to the promoter regions of these genes, activating RNA polymerase II, and facilitating the post-heat stress transcriptional response. The study further distinguished ENDU-2's role from that of HSF-1, DAF-12/FXR, and DAF-16/FOXO, which primarily regulate HSR during heat stress exposure. ENDU-2 was not required for survival under continuous heat stress but was essential for the long-term benefits of transient heat stress (Xu et al. 2023). These findings demonstrate that the protective effects of hormetic heat stress depend on coordinated transcriptional programs operating both during and after heat exposure, expanding the traditional focus of the field on mechanisms engaged during the immediate heat stress.

### Heat hormesis-induced transgenerational effects

Adopting the regimen of a single 1-h exposure at 35 °C on the day 1 of adulthood, Wan et al. observed remarkable transgenerational heat hormesis-induced lifespan extension, which persisted for up to 6 generations. They found that this heritable effect could be transmitted through both maternal and paternal lineages. Additionally, they discovered that proteostasis improvement previously associated with heat hormesis was also transgenerational, as evidenced by a reduction in accumulated α-synuclein protein in the Parkinson's disease model strain for 2 subsequent generations after heat stress priming. These findings suggest that both survival and fitness improvements conferred by heat hormesis can be passed to offspring. They identified HSF-1, DAF-16/FOXO, and DAF-12/FXR as essential for implementing the lifespan-extending mechanisms of heat hormesis, whereas only DAF-16/FOXO was required for transmitting these benefits to progeny. Furthermore, they demonstrated that transgenerational epigenetic inheritance of heat-induced longevity depends on both H3K9me3 histone modification and N6-mA DNA methylation. They found that elevated N6-mA levels co-localized with a reduction in H3K9me3 marks at heat stress response and autophagy-related genes, leading to persistent expression of these genes in the progeny, thereby promoting transgenerational protective benefits (Wan et al. 2021). Together, these findings indicate that DAF-16/FOXO, H3K9me3, and N6-mA modifications mediate the transgenerational inheritance of heat hormesis, extending its benefits beyond an individual's lifespan through epigenetic mechanisms.

Das et al. (2021) observed that heat priming day 1 adults for as little as 5 min at 34 °C resulted in enhanced thermotolerance in the next generation when challenged to a 37 °C heat shock, as well as reduced protein aggregation in a muscle-specific PolyQ

strain. Increasing the duration of the priming to 60 min resulted in this priming benefit being passed on for up to 2 generations. This heritable priming was attributed to germline HSF-1 recruitment of methyltransferase MET-2, which created repressive H3K9me2 marks at HSF-1 target genes, including *daf-2* (a repressor of *daf-16*). These chromatin modifications were retained in the progeny, and reduced *daf-16* expression, which was found to be essential for the enhanced thermotolerance, even more so than *hsf-1*. These results underline the importance of chromatin modifications to establishing a heritable hormetic response to heat priming.

## Repeated heat stress priming in adulthood

Repeated exposure to mild heat stress has been shown to induce greater hormesis effects. Olsen et al. applied a 4-h 33 °C regimen starting on day 1 of adulthood (the 4-d-old stage) and repeated the exposure every 4 d until day 13, while worms were maintained at 20 °C between exposure and throughout lifespan. Their findings revealed that more frequent exposures generally resulted in greater lifespan extension, although no linear additive effects were observed. Specifically, 4 exposures (on days 1, 5, 9, and 13 of adulthood) produced a greater lifespan extension than 3 (days 1, 5, and 9) or 2 exposures, and all were more effective than a single heat exposure. A similar trend was observed when using a 4-h exposure at 30 °C in the temperature-sensitive sterile mutant TJ1060 [fer-15(b26); spe-9(hc88)], with 3 exposures (on days 1, 5, and 9) yielding the strongest longevity benefits (Olsen et al. 2006). Possible mechanistic insights are discussed in the next section.

## Priming in late adulthood

With connections drawn between germline activity, longevity, and possibly thermotolerance, a key question remains of whether organisms can still experience hormetic benefits once they are no longer able to reproduce.

Olsen et al. (2006) showed that a single 4-h exposure to 33 °C on adulthood days 5 or 9, but not 13 increased lifespan. Interestingly, however, repeated exposures on days 5, 9, and 13 of adulthood also led to lifespan extension. Similarly, Kumsta et al. demonstrated that a single exposure to 36 °C for 1 h increased lifespan of day 3 adults but was less effective by day 5 and ineffective by day 7. Additionally, heat exposure on adulthood days 2, 3, and 5 but not 7, enhanced thermotolerance (Kumsta et al. 2017).

### Age-dependent decline in stress response inducibility

Mechanistically, both studies indicate that the decline in hormetic benefits with age parallels a reduced ability to induce protective stress response genes. Using an HSP-16.2::GFP reporter strain, Olsen et al. showed that hormetic heat treatment in young adults (primed at day 1 of adulthood, measured at day 3 of adulthood) dramatically induced HSP-16 expression, whereas induction was 100-fold lower in aged adults (primed at day 13 of adulthood, measured at day 15 of adulthood; Olsen et al. 2006). This finding is consistent with the work of Labbadia et al. (Labbadia and Morimoto 2015), which demonstrated that the HSR is rapidly repressed after the first day of adulthood. Olsen et al. further reported that repeated hormetic heat treatments (on day 1 and day 13 of adulthood) did not enhance HSP-16 induction in aged worms. The authors proposed that the loss of HSP inducibility in aged worms contributes both to the absence of late-life hormetic benefits and to the lack of additive effects from repeated treatments.

Kumsta et al. (2017) extended this observation, showing that HSP genes (*hsp-70, hsp-16.1*) and autophagy genes (*bec-1* and

*sqst-1*) remained inducible by hormetic heat stress from day 1 through 7 of the adulthood stage, whereas *atg-18* and *lgg-1* were only inducible on day 1. Together, these findings suggest that reduced inducibility of the heat-induced stress response pathway with age may underlie the declining efficacy of heat hormesis over time.

## Chronic heat exposure during development

### Thermosensitive TRP channel transduces signals to DAF-16/FOXO

Typically, *C. elegans* grown at higher temperature (25 °C) have a shorter lifespan compared to those maintained at 20 °C, with even greater lifespan extension maintained at 16 °C. However, Zhang et al. discovered that this lifespan trend is reversed when worms are exposed to higher or lower temperatures only during development and then returned to 20 °C upon reaching adulthood. Specifically, they identified the L1 to L2 larval stages as a critical window during which temperature exposure influences adult lifespan. Mechanistically, they found that temperature-mediated lifespan phenotypes are regulated by the thermosensitive TRP channel TRPA-1, which transduces signals to DAF-16/FOXO. DAF-16/FOXO then differentially regulates gene expression in a temperature-dependent manner, with distinct effects during development vs adulthood (Zhang et al. 2015). This study presents a different form of heat hormesis, demonstrating that chronic mild heat exposure during development promotes longevity.

### Histone acetylation-mediated sustained transcriptional activation

Using a developmental heat priming regimen (grown at 25 °C during development and then maintained at 15 °C upon reaching adulthood), Zhou et al. found that these worms exhibited enhanced resistance to both pathogen infection and oxidative stress, with the protective effects persisting for at least 6 d into adulthood. Specifically, worms primed at a higher developmental temperature displayed greater stress resistance by day 7 adulthood compared to those continuously maintained at lower temperatures. Mechanistically, they discovered that this enhanced stress resistance is driven by the persistent upregulation of immune and detoxification genes, which remained induced for 6 days post-heat exposure. This sustained upregulated expression was mediated by histone acetylation through the histone acetyltransferase CBP-1/p300, as evidenced by the fact that knocking down *cbp-1* abolished heat-induced stress resistance and the induction of defense genes. To further establish that epigenetic regulation via histone acetylation underlies the long-term stress resistance benefits, they performed ChIP-qPCR analysis, revealing increased occupancy of acetylated histones at the promoter regions of defense genes. This suggests that heat hormesis during development primes the activation of defense genes by promoting histone acetylation at their regulatory regions. Furthermore, they identified PMK-1/MAPK as an upstream regulator of histone acetylation, linking MAPK signaling to epigenetic modifications that mediate persistent stress resistance. In addition, they found that the SWI/SNF chromatin remodeling complex also contributes to hormetic heat stress-induced gene activation and stress defense phenotypes (Zhou et al. 2019). This study identifies epigenetic factors, including histone acetylation and chromatin remodeling, as key regulators of prolonged transcriptional alterations in heat hormesis, suggesting that epigenetic memory plays a crucial role in mediating the long-term physiological benefits of early-life heat exposure.

### Global improvement of proteostasis

Huang et al. provided a systematic proteomic analysis underlying heat hormesis. Rather than exposing worms to high temperatures throughout development, they showed that a 1-d exposure to 25 °C starting at the L4 stage was sufficient to extend lifespan and enhance resistance to multiple stressors, including heat shock, ultraviolet (UV), and oxidative stress, compared to worms maintained at 20 °C. These effects were regulated by transcription factors including HSF-1, DAF-16/FOXO, and HIF-1. Importantly, using quantitative proteomics, Huang et al. found that although elevated temperatures generally accelerate proteomic ageing, 1-d 25 °C priming during the pre-adult stage surprisingly improved proteostasis by reducing global protein production while enhancing protein chaperone maintenance and protein degradation. The temporal dynamics of the proteomic profile aligned with the observed lifespan outcomes: worms aged faster and lived shorter under continuous high temperatures but showed increased lifespan when heat priming was limited to a single day during a critical early-life window (Huang et al. 2023). This study highlights protein-level changes associated with the benefits of heat hormesis, though further studies are needed to establish their causal role in promoting longevity.

### TSP-1 protein multimerization for long-term protection

Jiang et al. demonstrated that 1-d exposure to 28 °C starting at the L4 larval stage also enhances thermotolerance and lifespan. This effect is regulated by Tetraspanin 1 (TSP-1), a conserved membrane protein that forms web-like structures following heat exposure, promoting protective mechanisms. Mechanistically, they found that TSP-1 transcription is activated during heat exposure by CBP-1, but not by HSF-1 or other common stress-response regulators. Interestingly, unlike previously reported mechanisms in which persistent gene induction contributes to long-term benefits, the mRNA levels of *tsp-1* are transient, returning to baseline around 24 h after shifting worms back to 20 °C. However, TSP-1 protein levels remain persistently upregulated for at least 48 h. This finding suggests that the transient transcriptional activation of *tsp-1*, mediated by CBP-1, leads to TSP-1 protein multimerization and the formation of web-like membrane structures, which serve as a form of cellular memory that confers long-term heat hormesis benefits (Jiang et al. 2024). This finding highlights protein perdurance as a mechanism for heat-induced stress resilience and longevity.

## Heat hormesis studies in the mammalian system highlight conserved mechanisms

It is well known that humans and other mammals can acclimate to stress-inducing temperatures (Cohen et al. 2007; Kassahn et al. 2009; Tian et al. 2011; Du et al. 2018). It is also evident that mild heat priming can imbue mammals with cross-tolerance to other stressors such as hypoxia and ischemic stress (Horowitz et al. 2004; Ely et al. 2014; Calabrese and Kozumbo 2021). While investigation into the molecular mechanisms behind heat hormesis has been less extensively studied in mammals, several studies have supported the findings of *C. elegans*-based research.

Many studies on mammalian heat hormesis come from the Michal Horowitz lab. They performed studies using a hormesis regimen in which 3-wk-old (juvenile) *Rattus norvegicus* were either kept at normothermic conditions (24 °C), primed at 34 °C for 30 d (long-term priming), or primed at 34 °C for 2 h (short-term priming) before being immediately subjected to either heat shock

(41–43 °C) for 2 h or cardiac ischemic/reperfusion (I/R) injury (Maloyan et al. 1999; Horowitz et al. 2004; Maloyan et al. 2005; Tetievsky et al. 2008, 2014; Assayag et al. 2010; Tetievsky and Horowitz 2010). They observed that long-term, but not short-term, priming reduced cardiac damage caused by heat shock or I/R injury (Tetievsky et al. 2008; Assayag et al. 2010). This effect was coupled with reduced apoptotic markers and an increase in the anti-apoptotic marker Bcl-X$_L$ in the long-term primed group (Assayag et al. 2010). Transcriptomic analysis similarly found that long-term primed rats had fewer transcripts of Bad (a pro-apoptotic marker) post-heat shock, and more rapid changes of transcript levels of HSP-70 and GST-P (an antioxidant gene) in response to I/R injury (Horowitz et al. 2004). Notably, long-term priming was observed to elevate expression of HSP-70 and HIF1A in both rats (Maloyan et al. 1999, 2005) and *C. elegans* (Treinin et al. 2003), suggesting evolutionarily conserved stress adaptation mechanisms.

In several studies, the researchers added an additional phase to the hormesis regimen in which, following 30-d priming, some rats were "de-acclimated" (DeAC) by returning to normothermic conditions for 1 or 2 months, followed by a "re-acclimation" (ReAC) at 34 °C for 2 d (Tetievsky et al. 2008, 2014; Tetievsky and Horowitz 2010). While DeAC animals experienced damage from heat shock and I/R injury similar to untreated control, ReAC animals rapidly re-gained the hormetic benefits associated with long-term priming, suggesting a hormetic memory persisting after at least 1 to 2 mo of normothermic recovery (Tetievsky et al. 2008). ChIP-qPCR analysis using this regimen revealed that 30-d priming caused elevated H4 acetylation and HSF-1 binding at the promoters of the *hsp-90* and *hsp-70* genes (Tetievsky and Horowitz 2010). Interestingly, HSF-1 binding at the *hsp-90* promoter is dynamic through DeAC/ReAC (declining during DeAC and regaining upon ReAC), whereas HSF-1 remained bound to the *hsp-70* promoter throughout DeAC and ReAC. These results suggest that HSP-70, whose protein level remains increased after 1 mo DeAC, is associated with hormetic memory, and that HSP-90, whose expression is more dynamic and is elevated during long-term priming and after heat shock only in primed animals, may directly impart hormetic stress resistance (Tetievsky et al. 2008; Tetievsky and Horowitz 2010). While the time scale and tissues involved obviously vary between mammals and *C. elegans,* these mammalian findings highlight parallel insights revealed by *C. elegans* studies: (i) HSF-1 and potentially HIF-1 mediated pathways are important for heat hormesis, (ii) persistent induction of HSPs underlies long-term benefits, (iii) epigenetic regulation such as histone modifications contribute to hormetic memory, and (iv) regimen optimization is critical for protective effects.

## Conclusions

Across the types of hormetic priming we have discussed—prolonged, short-term, chronic, repeated, or late-life—HSF-1 was broadly implicated. IIS-related factors, such as DAF-16/FOXO, as well as HIF-1, were observed to be essential for hormetic effects in multiple short-term and chronic priming regimens, suggesting a wide scope of importance. It is still unclear whether the hormetic mechanisms that impart lifespan extension overlap with those that impart increased thermotolerance. DAF-16/FOXO and DAF-12/FXR have been reported as essential for hormetic lifespan extension, but their requirement for thermotolerance appears to be more variable (Cypser and Johnson 2003; Kourtis et al. 2012; Zhang et al. 2015; Das et al. 2021; Wan et al. 2021; Huang et al. 2023). While these pathways are undoubtably key to hormesis, their ubiquity in the literature may be in part due to a bias of researchers toward choosing to measure their requirement. Unbiased screens identify additional regulators of hormesis that are vital in establishing hormesis through alternative pathways, although some molecular players are only engaged under particular hormesis conditions. For example, ENDU-2, one of these newly identified regulators, acts independently of HSF-1 and is suspected to recruit RNA Pol II to genes in the post-heat shock recovery period to provide hormetic resilience (Xu et al. 2023). However, while Xu et al. report ENDU-2 to be essential for priming-associated lifespan extension, cadmium resistance, and thermotolerance (Xu et al. 2023), it was dispensable for thermotolerance in another study (Chang et al. 2025). Histone acetylase CBP-1 and chromatin remodeling SWI/SNF are both essential for hormetic benefits associated with chronic mild temperature priming, though it remains to be seen whether they also act in response to shorter term, higher intensity heat stress (Zhou et al. 2019; Jiang et al. 2024). Even HSF-1, perhaps the most well-known mediator of heat shock response, has been reported to incur opposite effects on thermotolerance based on the age of worms at the time of heat stress (Kovács et al. 2024).

One vital consideration in heat hormesis experiments is the presence of a functional germline. HSF-1, ELT-2, and DPY-27 were found to vary in their requirement for hormetic thermotolerance depending on whether the strain was wildtype N2 or germline-less *glp-1(ts)* (Chang et al. 2025). Additionally, HSF-1 appears to be increasingly required in experiments in which germline proliferation was inhibited by FUDR, as discussed in section 1. There appears to be a more complex relationship between the germline and the soma, where heat priming elicits somewhat different molecular programs. Benefits to thermotolerance could act largely through canonical HSR in somatic cells, while benefits to longevity may be achieved through transient interruption of germline proliferation. Further investigation will be required to clarify this complexity.

Adaptations to mild heat stress occur via multiple mechanisms across various tissues and timescales, including the activity of protein-folding chaperones and induction of autophagy. A theme among heat hormesis studies is the transience of transcript-level regulation vs the persistence of proteins and chromatin modifications. Two studies have found that transcript-level changes induced by hormetic priming return to baseline in a matter of hours, despite lasting improvements in thermotolerance and longevity (Jiang et al. 2024; Chang et al. 2025). Chromatin changes, such as histone and DNA modifications are known to confer longer term "memory." Indeed, some chromatin modifications can last even for multiple generations after the initial heat priming (Wan et al. 2021). These modifications induce transcription of stress-response genes in offspring, leading to increased thermotolerance and lifespan.

## Future outlook

Despite growing interest in the field and emerging studies that reveal molecular insights underlying heat hormesis, significant gaps remain. In particular, the mechanisms engaged by specific regimens and their dependence on developmental stage or strain are not fully understood. For example, how do prolonged vs short priming during adulthood elicit distinct responses? How do interventions during development differ from those in adulthood, both phenotypically and mechanistically? And why does lifespan extension occur in some strains but not others under comparable conditions? An additional consideration is the spatiotemporal role of the molecular mediators of heat hormesis. Not only must we uncover factors involved in establishing and enacting hormetic protection, but also in what tissues they act in and in

what sequence. Establishing these facts would further set the stage for uncovering, for instance, whether hormesis occurs as a cascade across various tissues or cell-autonomously.

Moving forward, future studies would benefit from experimental designs that systematically control key regimen variables, such as temperature, timing, frequency, and genetic background, to enable more direct integration of findings across studies. Such standardization would strengthen cross-regimen comparisons and support the synthesis of insights relevant for guiding translational applications. In addition, examining whether long-term benefits come at the cost of trade-offs, such as impaired reproduction, will be important for understanding the full physiological impact of heat hormesis. Finally, comparing heat-induced hormesis with hormetic responses triggered by other stressors, such as dietary restriction, oxidative stress, or hypoxia, may reveal shared or distinct molecular pathways. Such insights could inform the development of targeted, multi-modal hormetic interventions to enhance resilience and promote healthy ageing.

## Acknowledgments

Thank you to all the authors whose work we have cited, and to all those who were not able to be covered in this review due to space limitation, for their dedication to the field. Thank you to Dr. David Lin, Dr. Andrew Grimson, and Dr. Praveen Sethupathy for their feedback. Figure 1 was created using BioRender.com. Thank you to Dr. Amaresh Chaturbedi for feedback and continuous support. Portions of this text also appear in a dissertation titled "Unraveling the Molecular Basis of Stress Adaptation and Longevity in C. elegans Through the Lens of Heat Hormesis" completed by H-Y.C. to fulfill the requirements of the Cornell University Biomedical and Biological Sciences Ph.D. program (Chang 2025).

## Funding

This work was funded by the National Institutes of Health AG024425 (to S.S.L.) and the Ministry of Education, Taiwan (to H-Y.C.).

## Author contributions

H-Y.C.: Writing—original draft, editing, and integration; C.L.H.: Writing—mammalian and conclusions sections, revisions, editing; S.S.L.: Funding acquisition, writing—review, and editing.

## Declaration of generative AI and AI-assisted technologies

During the preparation of this work, the authors used ChatGPT-4o in order to improve language and readability. After using this tool/service, the authors reviewed and edited the content as needed and take full responsibility for the content of the publication.

## Conflicts of interest

None declared.

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

*Editor: C. Phillips*