## [Peer Review File · Genetics]

Molecular insights into diverse heat hormesis regimens in *Caenorhabditis elegans*

Siu Lee, Hsin-Yun Chang, and Charles Heinke

NOTE: The reviews and decision letters are unedited and appear as submitted by the reviewers.

In extremely rare instances and as determined by a Senior Editor or the EIC, portions of a review may be redacted. If a review is signed, the reviewer has agreed to no longer remain anonymous.

The review history appears in chronological order.

Review Timeline:

Submission Date:	2025-09-10
Editorial Decision:	2025-10-15
Revision Received:	2026-01-06
Accepted:	2026-01-07

October 15, 2025

RE: GENETICS-2025-308582

Dear Dr. Lee:

I am pleased to accept your manuscript titled "Molecular insights into diverse heat hormesis regimens in *Caenorhabditis elegans*" for publication in GENETICS, pending minor revision.

Please submit your revision along with a brief description of how you modified the manuscript in response to the reviewers' concerns and suggestions (which can be viewed at the bottom of this email. Most important are to include a stronger, integrative conclusion that clearly synthesizes shared and regimen-specific molecular mechanisms and to ensure that you are clearly defining key terms, consistently describing how experiments were done, and giving readers enough methodological detail to understand and compare findings across studies.. I expect you should be able to submit a revised manuscript within 30 days. A suitably revised manuscript will be acceptable for publication; I don't expect to send it out for review.

Thank you for submitting this story to Genetics.

Sincerely,

Carolyn Phillips
Associate Editor
GENETICS

Approved by:
Julie Claycomb
Senior Editor
GENETICS

Reviewer comments:

Reviewer #1 :

This is an excellent and timely review that provides a comprehensive synthesis of the literature on heat hormesis in *C. elegans*. The manuscript is well written, logically structured, and successfully integrates work across different experimental regimens. I particularly appreciate the inclusion of mammalian studies, which effectively highlight the evolutionary conservation of hormetic mechanisms and expand the translational relevance of the topic.

Overall, this review will serve as a valuable reference for the field. I have a few suggestions aimed at strengthening the synthesis and deepening some mechanistic interpretations.

General Comments

1. The conclusion would benefit from a dedicated summary paragraph that explicitly compares the molecular pathways across regimens. For instance, the Heat Shock Response (HSR) and IIS appear to be unifying mechanisms across nearly all paradigms, whereas autophagy and epigenetic regulation have been examined in fewer contexts. Highlighting which mechanisms are broadly required versus regimen-specific would give readers a clearer sense of hierarchy and integration among molecular processes. Moreover, how the different pathways (e.g. HSR/HSF-1, DAF-16/IIS, autophagy, and epigenetic regulation) interact or converge on shared downstream outcomes such as proteostasis or cellular resilience, should be better emphasized in the conclusions. The link between acute stress responses (e.g., transient HSP induction) and long-term or even transgenerational benefits is also not clearly articulated, despite the intriguing mention of "molecular memory." A stronger synthesis could connect short-term transcriptional or proteostatic responses with longer-term chromatin and epigenetic adaptations under a unifying concept of stress-induced cellular memory.
2. Spatiotemporal regulation: An important open question concerns when and where these molecular players act to mediate hormetic benefits. A short reflection on this knowledge gap in the Outlook section would strengthen the forward-looking aspect of the paper and emphasize the need for spatiotemporal resolution in future studies.

Specific Textual Comments

1. Line 147 on: The section describing the *hsf-1(sy441)* mutant is clear and informative. However, additional context should be provided noting that *hsf-1* knockdown by RNAi has also been tested in some hormetic regimens and impacts certain aspects of

stress resilience. Including this would round out the discussion and acknowledge the complementary experimental approaches used to dissect HSF-1 function.

2. Lines 175-176: Please specify the recovery period used in the referenced experiment.

3. Line 179 on, MARS-1, SNPC-4, FOS-1, and DPY-27 Section: This paragraph ends somewhat abruptly. These factors represent intriguing new regulatory layers, translation, small-RNA pathways, transcriptional control, and chromatin architecture. A brief interpretive statement suggesting how these might be induced by heat shock and contribute to hormetic adaptation (for example, by maintaining transcriptional competence or genome organization during recovery) would make the section more engaging and biologically informative.

4. Lines 213-220: Differentiating Lifespan vs. Thermal-Resistance. The text distinguishes between thermal resistance and lifespan extension but does not explore why they may arise through partially distinct mechanisms. Adding a short mechanistic interpretation here would strengthen the discussion.

5. Lines 264 on: It might be worth including that that hormetic heat stress reduced PolyQ by increasing the mobility of neuronal PolyQ aggregates in FRAP assays and increased the soluble portion and decreased the insoluble portion of PolyQ proteins in DDE assays (in an selective autophagy receptor dependent manner (sqst-1/p62) (Kumsta, Nature Communication, 2019) Please also note that heat hormesis increased lifespan in both intestine- and neuron-specific PolyQ models.

6. Line 321: Please clarify that lifespan was extended even after priming at day 5 of adulthood, albeit to a lesser extent.

Reviewer #2 :

Heat hormesis is described as a phenomenon where exposure to a mild heat stress, followed by a recovery period, promotes resilience to a future heat stress. Many studies have been performed investigating heat hormesis using *C. elegans* under various experimental regimens, and this review aims to organize these results according to the duration, timing, and intensity of the heat stress. This review fills a gap in the synthesis of knowledge regarding hormesis and would be useful those with an interest in stress biology.

Major concerns

Figure 1: The figure is not explained well by the legend. There is no blue dashed line as described. It is also unclear what the significance of the orange line and the black arrow are between the two figures. Either the figure or the legend should be revised.

Line 121: "WT and long-lived strains show increased lifespan..." This title seems redundant as written. The section titles could be clarified throughout to state the conclusion of the relevant studies.

Line 151 and throughout: it would be helpful to define how "thermal resistance" is measured by each study. It is survival, reproduction, or longevity? More clarity in the description of the studies would be helpful.

Line 208 and others: There are multiple studies that use temperature-sensitive sterile strains described in this review. It would improve this review to include a section that remarks on the requirement of a germline in thermal resistance. In "Future Outlook", it would also be useful to have some overall conclusions about the mechanisms of heat hormesis in addition to the future directions. The comments regarding a germline could be included in this section. As is, the review reads as an individual's paper notes, and a conclusion section that synthesizes themes of the results would be helpful to make the review more cohesive.

Line 402: if including the section on mammalian studies, it should be mentioned in the abstract and introduction. Currently, the expectation is that only *C. elegans* studies would be included.

Minor concerns

Line 88: the text references a review from "Patrick and Johnson", but the citations include more than that review.

Lines 100-109: consider combining lines 108-109 with 100-102. As is, lines 100-102 is repetitive with the list of sections below, and it would be more useful to list how the regimens differ as listed in lines 108-109.

Table 1 and throughout manuscript: some genes are italicized in the table and some not. They should all be italicized to conform with the *C. elegans* standards.

Table 1: it is unclear what "HSF-1 -> HSP-16.1" indicates.

Table 1: "peorid" is misspelled multiple times

Line 124: it would be useful to put a statement of why ageing mutants were tested in thermotolerance assays

Line 173: does "RNA expression" refer to mRNA only or also noncoding RNAs?

Line 180-182: this sentence is confusing

Line 214: "and" is italicized

Reviewer comments:

Reviewer #1 :

This is an excellent and timely review that provides a comprehensive synthesis of the literature on heat hormesis in *C. elegans*. The manuscript is well written, logically structured, and successfully integrates work across different experimental regimens. I particularly appreciate the inclusion of mammalian studies, which effectively highlight the evolutionary conservation of hormetic mechanisms and expand the translational relevance of the topic.

Overall, this review will serve as a valuable reference for the field. I have a few suggestions aimed at strengthening the synthesis and deepening some mechanistic interpretations.

General Comments

1. The conclusion would benefit from a dedicated summary paragraph that explicitly compares the molecular pathways across regimens. For instance, the Heat Shock Response (HSR) and IIS appear to be unifying mechanisms across nearly all paradigms, whereas autophagy and epigenetic regulation have been examined in fewer contexts. Highlighting which mechanisms are broadly required versus regimen-specific would give readers a clearer sense of hierarchy and integration among molecular processes. Moreover, how the different pathways (e.g. HSR/HSF-1, DAF-16/IIS, autophagy, and epigenetic regulation) interact or converge on shared downstream outcomes such as proteostasis or cellular resilience, should be better emphasized in the conclusions. The link between acute stress responses (e.g., transient HSP induction) and long-term or even transgenerational benefits is also not clearly articulated, despite the intriguing mention of "molecular memory." A stronger synthesis could connect short-term transcriptional or proteostatic responses with longer-term chromatin and epigenetic adaptations under a unifying concept of stress-induced cellular memory.

We thank the reviewer for their comments. We have added a "Conclusion" section tying together the themes of the *C. elegans* section of the review. In this paragraph we discuss factors that are shared and unique across the regimen types. We also discuss the various mechanisms by which hormetic benefits are imparted by various factors, and how these may be responsible for different aspects of the time-scale of hormetic benefits.

2. Spatiotemporal regulation: An important open question concerns when and where these molecular players act to mediate hormetic benefits. A short reflection on this knowledge gap in the Outlook section would strengthen the forward-looking aspect of the paper and emphasize the need for spatiotemporal resolution in future studies.

We now address this question at the end of the second paragraph in the Future Outlook.

Specific Textual Comments

1. Line 147 on: The section describing the hsf-1(sy441) mutant is clear and informative. However, additional context should be provided noting that hsf-1 knockdown by RNAi has also been tested in some hormetic regimens and impacts certain aspects of stress resilience. Including this would round out the discussion and acknowledge the complementary experimental approaches used to dissect HSF-1 function.

We have included mention of hsf-1 RNAi producing similar results to the mutant strain in two studies.

2. Lines 175-176: Please specify the recovery period used in the referenced experiment.

We have added this information.

3. Line 179 on, MARS-1, SNPC-4, FOS-1, and DPY-27 Section: This paragraph ends somewhat abruptly. These factors represent intriguing new regulatory layers, translation, small-RNA pathways, transcriptional control, and chromatin architecture. A brief interpretive statement suggesting how these might be induced by heat shock and contribute to hormetic adaptation (for example, by maintaining transcriptional competence or genome organization during recovery) would make the section more engaging and biologically informative.

We agree with the reviewer that those new factors are exciting. We have included a brief summary regarding the cellular processes and pathways that these factors point to.

4. Lines 213-220: Differentiating Lifespan vs. Thermal-Resistance. The text distinguishes between thermal resistance and lifespan extension but does not explore why they may arise through partially distinct mechanisms. Adding a short mechanistic interpretation here would strengthen the discussion.

We hypothesize that longevity extension may be primarily due to transient disruption of germline proliferation, while heat shock response pathways in somatic cells are more responsible for increases in thermotolerance. We have included this interpretation to the conclusion and “Germline-less and long-lived glp-1(ts) mutants do not exhibit further lifespan extension upon priming” section.

5. Lines 264 on: It might be worth including that that hormetic heat stress reduced PolyQ by increasing the mobility of neuronal PolyQ aggregates in FRAP assays and increased the soluble portion and decreased the insoluble portion of PolyQ proteins in DDE assays (in an selective autophagy receptor dependent manner (sqst-1/p62) (Kumsta, Nature Communication, 2019) Please also note that heat hormesis increased lifespan in both intestine- and neuron-specific PolyQ models.

We have included a reference to Kumsta et al, 2019 highlighting the importance of sqst-1 in priming-induced autophagy. We have also added mention of the lifespan benefit in the neuron-specific PolyQ model.

6. Line 321: Please clarify that lifespan was extended even after priming at day 5 of adulthood, albeit to a lesser extent.

We have added this detail.

Reviewer #2 :

Heat hormesis is described as a phenomenon where exposure to a mild heat stress, followed by a recovery period, promotes resilience to a future heat stress. Many studies have been performed investigating heat hormesis using *C. elegans* under various experimental regimens, and this review aims to organize these results according to the duration, timing, and intensity of the heat stress. This review fills a gap in the synthesis of knowledge regarding hormesis and would be useful those with an interest in stress biology.

Major concerns

Figure 1: The figure is not explained well by the legend. There is no blue dashed line as described. It is also unclear what the significance of the orange line and the black arrow are between the two figures. Either the figure or the legend should be revised.

We thank the reviewer for their feedback. We have switched the figure to the proper version that includes the features described in the legend.

Line 121: "WT and long-lived strains show increased lifespan..." This title seems redundant as written. The section titles could be clarified throughout to state the conclusion of the relevant studies.

We have changed this title to: "Priming duration threshold altered in long-lived mutant". We have also changed several other section titles to be more descriptive.

Line 151 and throughout: it would be helpful to define how "thermal resistance" is measured by each study. It is survival, reproduction, or longevity? More clarity in the description of the studies would be helpful.

We have added a definition of thermal-resistance in section 1 as “an increase in survival after an intense heat stress”.

Line 208 and others: There are multiple studies that use temperature-sensitive sterile strains described in this review. It would improve this review to include a section that remarks on the requirement of a germline in thermal resistance. In "Future Outlook", it would also be useful to have some overall conclusions about the mechanisms of heat hormesis in addition to the future directions. The comments regarding a germline could be included in this section. As is, the review reads as an individual's paper notes, and a conclusion section that synthesizes themes of the results would be helpful to make the review more cohesive.

We have added a discussion of the germline in the “Conclusions” section. We have also included an observation in “A conditional requirement of HSF-1” that HSF-1 appears to vary in its requirement for hormesis-induced benefits based on the presence of a functional germline, and that this may be due to its pro-proliferative effect in germline precursor cells.

Line 402: if including the section on mammalian studies, it should be mentioned in the abstract and introduction. Currently, the expectation is that only *C. elegans* studies would be included.

We have added this mention to the abstract and introduction.

Minor concerns

Line 88: the text references a review from "Patrick and Johnson", but the citations include more than that review.

We have corrected this.

Lines 100-109: consider combining lines 108-109 with 100-102. As is, lines 100-102 is repetitive with the list of sections below, and it would be more useful to list how the regimens differ as listed in lines 108-109.

We have made this more concise.

Table 1 and throughout manuscript: some genes are italicized in the table and some not. They should all be italicized to conform with the *C. elegans* standards.

Table 1: it is unclear what "HSF-1 -> HSP-16.1" indicates.

Table 1: "peorid" is misspelled multiple times

These have been corrected.

Line 124: it would be useful to put a statement of why ageing mutants were tested in thermotolerance assays

We have added the context that long-lived mutants were noticed to have innately increased thermal-resistance, suggesting a link between longevity and thermal-resistance.

Line 173: does "RNA expression" refer to mRNA only or also noncoding RNAs?

We have specified this to mean mRNA, as polyA selection was used.

Line 180-182: this sentence is confusing

This sentence has been reworded for clarity.

Line 214: "and" is italicized

This has been corrected.

January 7, 2026

RE: GENETICS-2025-308582R1

Dr. Siu Sylvia Lee
Cornell University
Molecular Biology and Genetics
339 Biotechnology Building, Cornell University
Ithaca, New York 14853

Dear Dr. Lee:

Congratulations, your Review titled "Molecular insights into diverse heat hormesis regimens in *Caenorhabditis elegans*" is accepted for publication in GENETICS! Many thanks for contributing to GENETICS.

To Proceed to Publication:

1. Format your article according to GENETICS style: <https://academic.oup.com/genetics/pages/author-guidelines>
2. Ensure that you comply with data and community resource citation guidelines: <https://academic.oup.com/genetics/pages/author-guidelines#section-5-9-2>
3. Upload your final files at <https://genetics.msubmit.net>
4. Add oupsupport@scipris.com and genetics.oup@novatechset.com (or the domains @scipris.com and @novatechset.com) to your email program's "safe senders" list. You will be contacted by both at various points during the production process.

Notes:

- We invite you to submit an original color figure related to your paper for consideration as cover art. Please email your submission to the editorial office or upload it with your final files. You can submit a small-sized image for evaluation, and if selected, the final image must be a TIFF file 2513px wide by 3263px high (8.375 by 10.875 inches; resolution of 600ppi). Please avoid graphs and small type.

- After files are sent to Oxford University Press we use SciPris to manage article licensing and payment. If you do not have a SciPris account, you will receive an email from no-reply@scipris.com to sign up to use Oxford University Press' author portal. After logging in, follow the online instructions to sign your licence. It is important that you select the Standard License to Publish so that the GSA will be billed for the page charges (Open Access is not covered by the GSA).

If you have any questions or encounter any problems while uploading your accepted manuscript files, please email the editorial office at sourcefiles@thegsajournals.org.

Congratulations and happy new year!

Carolyn Phillips
Associate Editor
GENETICS

Approved by:
Julie Claycomb
Senior Editor
GENETICS